# HYDRA:
# PRESERVING ENSEMBLE DIVERSITY FOR
# MODEL DISTILLATION

## ABSTRACT

Ensembles of models have been empirically shown to improve predictive performance and to yield robust measures of uncertainty. However, they are expensive in computation and memory. Therefore, recent research has focused on *distilling* ensembles into a single compact model, reducing the computational and memory burden of the ensemble while trying to preserve its predictive behavior. Most existing distillation formulations summarize the ensemble by capturing its *average* predictions. As a result, the *diversity* of the ensemble predictions, stemming from each individual member, is lost. Thus, the distilled model cannot provide a measure of uncertainty comparable to that of the original ensemble. To retain more faithfully the diversity of the ensemble, we propose a distillation method based on a single multi-headed neural network, which we refer to as *Hydra*. The shared body network learns a joint feature representation that enables each head to capture the predictive behavior of each ensemble member. We demonstrate that with a slight increase in parameter count, Hydra improves distillation performance on classification and regression settings while capturing the uncertainty behaviour of the original ensemble over both in-domain and out-of-distribution tasks.

## 1 INTRODUCTION

Deep neural networks have achieved impressive performance, however, they tend to make over-confident predictions and poorly quantify uncertainty (Lakshminarayanan et al., 2017). It has been demonstrated that ensembles of models improve predictive performance and offer higher quality uncertainty quantification (Dietterich, 2000; Lakshminarayanan et al., 2017; Ovadia et al., 2019). A fundamental limitation of ensembles is the cost of computation and memory at evaluation time. A popular solution is to distill an ensemble of models into a single compact network by attempting to match the average predictions of the original ensemble. This idea goes back to the foundational work of Hinton et al. (2015), itself inspired by earlier ideas developed by Bucilu et al. (2006). While this process has led to simple and well-performing algorithms, it fails to take into account the intrinsic diversity of the predictions of the ensemble, as represented by the individual predictions of each of its members. In particular, this diversity is all the more important in tasks that hinge on the uncertainty output of the ensemble, e.g., in out-of-distribution scenarios (Lakshminarayanan et al., 2017; Ovadia et al., 2019). Similarly, by losing the diversity of the ensemble, this simple form of distillation makes it impossible to estimate measures of uncertainty such as model uncertainty (Depeweg et al., 2017; Malinin et al., 2019). Proper uncertainty quantification is especially crucial for safety-related tasks and applications. To overcome this limitation, Malinin et al. (2019) proposed to model the entire distribution of an ensemble using a Dirichlet distribution parametrized by a neural network, referred to as a prior network (Malinin & Gales, 2018). However, this imposes a strong parametric assumption on the distillation process.

Inspired by multi-headed architectures already widely applied in various applications (Szegedy et al., 2015; Sercu et al., 2016; Osband et al., 2016; Song & Chai, 2018), we propose a multi-headed model to distill ensembles. Our multi-headed approach—which we name *Hydra*—can be seen as an interpolation between the full ensemble of models and the knowledge distillation proposed by Hinton et al. (2015). Our distillation model is comprised of (1) a single *body* and (2) as many *heads* as there are members in the original ensemble. Each head is assigned to an ensemble member

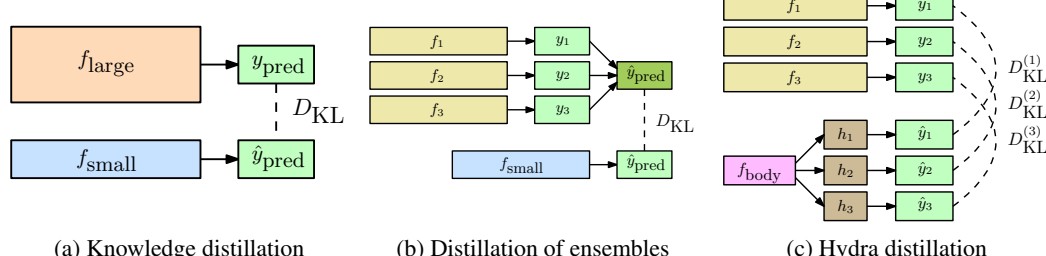

(a) Knowledge distillation  (b) Distillation of ensembles  (c) Hydra distillation

Figure 1: Existing distillation methods compared to Hydra. *Knowledge distillation*, (Hinton et al., 2015), trains a distillation network to imitate the prediction of a larger network. Applying knowledge distillation to ensemble models Hinton et al. (2015) train a network to imitate the average ensemble prediction. *Hydra* instead learns to distill the individual predictions of each ensemble member into separate light-weight head models while amortizing the computation through a shared heavy-weight body network. This retains the diversity of ensemble member predictions which is otherwise lost in knowledge distillation.

and tries to mimic the *individual predictions* of this ensemble member, as illustrated in Figure 1. The heads share the same body network whose role is to provide a common feature representation. The design of the body and the heads makes it possible to trade off the computational and memory efficiency against the fidelity with which the diversity of the ensemble is retained. An illustration of the common knowledge distillation and ensemble distillation as well as Hydra is shown in Figure 1 and a detailed methodology description is found in Section 2. While the choices of the body and head architectures may appear like complex new hyperparameters we introduce, we will see in the experiments that we get good results by simply taking the $N - k, 0 \leq k \ll N$, layers of the original ensemble members for the body and duplicating the layer for the heads.

**Summary of contributions.** Firstly, we present a multi-headed approach for the ensemble knowledge distillation. The shared component keeps the model computationally and memory efficient while diversity is captured through the heads matching the individual ensemble members. Secondly, we show through experimental evaluation that Hydra outperforms existing distillation methods for both classification and regression tasks with respect to predictive test performance. Lastly, we investigate Hydra's behaviour in terms of in-domain and out-of-distribution data and demonstrate that Hydra comes closest to the ensemble behaviour in comparison to existing distillation methods.

**Novelty and significance.** Ensembles of models have successfully improved predictive performance and yielded robust measures of uncertainty. However, existing distillation methods do not retain the diversity of the ensemble (beyond its average predictive behavior), or need to make strong parametric assumptions that are not applicable in regression settings. To the best of our knowledge, our approach is the first to employ a multi-headed architecture in the context of ensemble distillation. It is simple to implement, does not make the strong parametric assumptions, requires few modifications to the distilled ensemble model and works well in practice, thereby making it attractive to apply to a wide range of ensemble models and tasks.

## 2 HYDRA: A MULTI-HEADED APPROACH

With a focus on offline distillation, our goal is to train a student network to match the predictive distribution of the teacher models, which is an ensemble of (deep) neural networks. Formally, given a dataset $\mathcal{D} = \{(\mathbf{x}^{(i)}, \mathbf{y}^{(i)})\}_{i=1}^{N}$, we consider an ensemble of M models' parameters $\theta_{\text{ens}} = \{\theta_{\text{ens},m}\}_{m=1}^{M}$ and prediction outputs $\{p(\mathbf{y}^{(i)}|\mathbf{x}^{(i)}; \theta_{\text{ens},m})\}_{m=1}^{M}$. For simplicity, a single data instance pair will be referred to as $(\mathbf{x}, \mathbf{y})$.

In (Hinton et al., 2015; Balan et al., 2015) distilling an ensemble of models into a single neural network is achieved by minimizing the Kullback-Leibler (KL) divergence between the student's predictive distribution $\{p(\mathbf{y}|\mathbf{x}; \theta_{\text{distill}})\}$ and the expected predictive distribution of an ensemble:

$$\mathcal{L}(\theta_{\text{ens}}, \theta_{\text{distill}}) = \mathbb{E}_x\left[\text{KL}\left(\frac{1}{M} \sum_{m=1}^{M} (p(\mathbf{y}|\mathbf{x}; \theta_{\text{ens},m})) \parallel p(\mathbf{y}|\mathbf{x}; \theta_{\text{distill}})\right)\right] \tag{1}$$

Hydra builds upon the approach of knowledge distillation and extends it to a multi-headed student model. Hydra is defined as a (deep) neural network with a single body and $M$ heads. For distillation, Hydra has as many heads as there are ensemble members. The distillation model is parametrized by $\theta_{\text{hydra}} = \{\theta_{\text{body}}, \{\theta_{\text{head},m}\}_{m=1}^{M}\}$, i.e. the body, $\theta_{\text{body}}$, is shared among all heads $\{\theta_{\text{head},m}\}_{m=1}^{M}$.

In terms of number of parameters, we assume the heads to be much lighter than the shared part, so that the distillation is still meaningful. In practice, we use $N - k, 0 \leq k \ll N$, layers of the original ensemble member architecture and the original final layer(s) as head. The objective is to minimize the average KL divergence between each head $m$ and corresponding ensemble member $m$. We differentiate between two tasks, classification and regression.

**Classification.** For classification tasks, the ensemble of models has access to $\mathcal{D}$ during training, with each x belonging to one of $C$ classes, i.e., $\text{y} \in \{1, 2, \ldots, C\}$. Assuming $\text{z}_m = f_{\theta_{\text{head},m}}(f_{\theta_{\text{body}}}(\text{x}))$ correspond to the logits, also termed unnormalised log probabilities, of head $m$, the categorical distribution over class labels for a sample $x$ over a class $c$ is computed as:

$$p(c|\text{x}; \theta_{\text{hydra},m}) = p(c|\text{z}_m) = \frac{\exp(\text{z}_{m,c}/T)}{\sum_{j=1}^{C} \exp(\text{z}_{m,j}/T)}, \tag{2}$$

where $T$ is a temperature re-scaling the logits. As discussed in (Hinton et al., 2015; Malinin et al., 2019) the distribution of the teacher network is often "sharp", which can limit the common support between the output distribution of the model and the target empirical distribution. Mimimizing KL-divergence between distributions with limited non-zero common support is known to be particularly difficult. To alleviate this issue, we follow the common practice (Hinton et al., 2015; Song & Chai, 2018; Lan et al., 2018) to use temperature to "heat up" both distributions and increase common support during training. At evaluation, $T$ is set to 1. The soft probability distributions at a temperature of T are used to match the teacher ensemble of models by minimizing the average KL divergence between each head $m$ and ensemble model $m$:

$$\mathcal{L}(\text{x}, \text{y}; \theta_{\text{hydra}}, \theta_{\text{ens}}) = \frac{1}{M} \sum_{m=1}^{M} \text{KL}\left(p(\text{y}|\text{x}; \theta_{\text{ens},m}) \parallel p(\text{y}|\text{x}; \theta_{\text{body}}, \theta_{\text{head},m})\right). \tag{3}$$

Compared to the objective of knowledge distillation (1), we can observe that the average over the ensemble members is pulled out of the KL. Ignoring the constant entropy terms, this objective is reduced to standard cross entropy loss:

$$\mathcal{L}(\text{x}, \text{y}; \theta_{\text{head}}, \theta_{\text{ens}}) = -\frac{T^2}{M} \sum_{m=1}^{M} p(\text{y}|\text{x}; \theta_{\text{ens},m}) \log p(\text{y}|\text{x}; \theta_{\text{body}}, \theta_{\text{head},m}). \tag{4}$$

We scale our objective by $T^2$ as the gradient magnitudes produced by the soft targets are scaled by $1/T^2$. By multiplying the loss term by a factor of $T^2$ we ensure that the relative contributions to additional regularization losses remain roughly unchanged (Song & Chai, 2018; Lan et al., 2018).

**Regression.** We focus on heteroscedastic regression tasks where each ensemble member $m$ outputs a mean $\mu_m(x)$ and $\sigma_m^2(x)$ given an input x.[1] The output is modeled as $p(\text{y}|x, \theta_m) = \mathcal{N}(\mu_m(x), \sigma_m^2(x))$ for a given head $m$ and the ensemble of models are trained by minimizing the negative log-likelihood. Traditional knowledge distillation matches a single Gaussian ("student") outputting $\mu_{\text{distill}}(x)$ and $\sigma_{\text{distill}}^2(x)$ to a mixture of Gaussians (a "teacher" ensemble):

$$\mathcal{L}(\text{x}, \text{y}; \theta_{\text{ens}}, \theta_{\text{distill}}) = \text{KL}\left(\frac{1}{M} \sum_{m=1}^{M} \mathcal{N}(\mu_m(x), \sigma_m^{2,(i)}(x)) \parallel \mathcal{N}(\mu_{\text{distill}}(x), \sigma_{\text{distill}}^2(x))\right) \tag{5}$$

With Hydra, each head $m$ outputs a mean $\mu_{\text{hydra},m}(x)$ and variance $\sigma_{\text{hydra},m}^2(x)$ and optimizes the KL divergence between each head output and corresponding ensemble member output:

---

[1]In our concrete implementation, our neural network outputs the mean $\mu_m(x)$ and log standard deviation $\log(\sigma_m^2(x))$ which we thereafter exponentiate.

$$\mathcal{L}(\mathbf{x}, \mathbf{y}; \theta_{\text{ens}}, \theta_{\text{hydra}}) = \frac{1}{M} \sum_{m=1}^{M} \text{KL}\left( \mathcal{N}(\mu_m(x), \sigma_m^{2,(i)}(x)) \parallel \mathcal{N}(\mu_{\text{hydra},m}(x), \sigma_{\text{hydra},m}^2(x)) \right),$$

$$\underset{\text{(omitting constants)}}{\propto} \quad -\frac{1}{M} \sum_{m=1}^{M} \int \mathcal{N}(\mathbf{y}; \mu_m(x), \sigma_m^{2,(i)}(x)) \log \mathcal{N}(\mathbf{y}; \mu_{\text{distill}}, \sigma_{\text{distill}}^2) \, dy$$

$$= \quad -\frac{1}{M} \sum_{m=1}^{M} \frac{\sigma_m^2(x) + (\mu_m(x) - \mu_{\text{distill}}(x))^2}{2\sigma_{\text{distill}}^2(x)} + \frac{1}{2} \log(2\pi\sigma_{\text{distill}}^2(x))$$

where the final line uses the fact that each KL term has an analytical solution.

**Training with multi-head growth.** Hydra is trained in two phases. In the first phase, Hydra mimics knowledge distillation in that it is trained until convergence with a single head—the "Hinton head"—to match the average predictions of the ensemble. Hydra is then extended by $M - 1$ heads, all of which initialized with the parameter values of the "Hinton head". The resulting $M$ heads are finally further trained to match the individual predictions of the $M$ ensemble members (according to objective (3)). In practice, we sometimes experienced difficulties for Hydra to converge in absence of this initialization scheme, and for the cases where different initialization worked, this two-phase training scheme typically led to overall quicker convergence.

## 3 EVALUATION

In this section, we demonstrate that Hydra not only best matches the behavior of the teacher ensemble in terms of uncertainty quantification but also improves the predictive performance compared to existing distillation approaches, over both classification and regression tasks.

**Datasets.** For visualizing and explaining model uncertainty in Subsection 3.1 we used a spiral toy dataset. For classification, we used two datasets: MNIST and CIFAR-10. For evaluating MNIST, we use its test set as well as increasingly shifted data (increasingly rotated or horizontally translated images) and Fashion-MNIST. For CIFAR-10, we report performance of the test set, the cyclic translated test set and 80 different corrupted test sets as well as SVHN. The pre-processing for MNIST and CIFAR-10, as well as the generation schemes of the corrupted images, are taken from Ovadia et al. (2019). For regression, we conducted experiments on the standard regression datasets from the UCI repository (Asuncion & Newman, 2007), following the protocol of Bui et al. (2016).

**Ensemble setup.** For the toy dataset, we trained 10 ensemble models, each of which consists of a multi-layer perceptron (MLP) with two hidden layers of 100 units each. For MNIST and CIFAR, we each trained ensembles of 50 models. For ensemble training we used different architectures for CIFAR-10 and MNIST. We used an MLP architecture for MNIST and the ResNet-20 V1 architecture for CIFAR-10. For additional details for MNIST and CIFAR-10 models, we refer to (Ovadia et al., 2019) and their open-source code[2]. For all regression tasks, the same model is optimized: an MLP with a single hidden layer of 50 units with softplus activation for each dataset, except for case of the larger protein structure dataset, `prot` where 100 units were used (following Bui et al. (2016)).

**Distillation setup.** We compare our work with two core distillation approaches, Knowledge Distillation (Hinton et al., 2015) and Prior Networks (Malinin et al., 2019; Malinin & Gales, 2018). All baseline models have the same architecture as the ensemble for distillation. For MNIST, Hydra uses the original ensemble member architecture and adds an MLP with two hidden layers of 100 units each as head. For CIFAR-10, the original Resnet20 V1 model without the last residual block was used as body. The final distillation model reported here has one residual block per head.

**Evaluation metrics.** We report all evaluation metrics on the test set based on the best validation loss from training. For both classification and regression, we evaluate the negative log likelihood (NLL) as well as Brier score which depends on the predictive uncertainty and model uncertainty (MU). NLL is a proper scoring rule and a popular metric for evaluating predictive uncertainty (Ovadia

---

[2]https://github.com/google-research/google-research/tree/master/uq_benchmark_2019

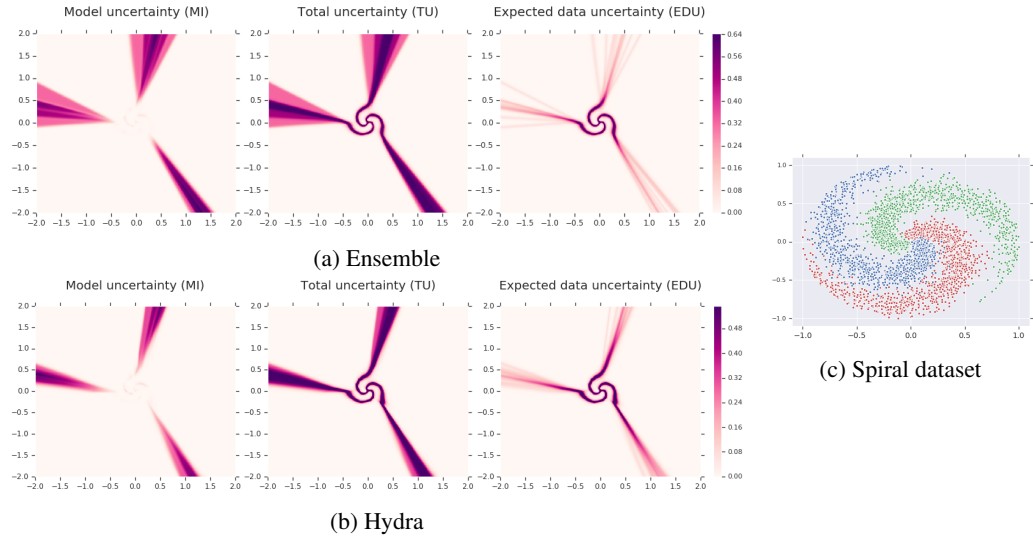

Figure 2: Model uncertainty, total uncertainty and expected data uncertainty applied on in-domain and out-of-distribution data for both (a) ensemble and (b). The original dataset is visualized (c), where each color corresponds to a single class.

| Dataset | Knowledge distillation (Hinton et al., 2015) | Prior Networks (Malinin et al., 2019) | Hydra |
|---|---|---|---|
| MNIST (test) | N/A | $2.60 \cdot 10^{-5}$ $_{\pm 3.34 \cdot 10^{-5}}$ | $\mathbf{3.11 \cdot 10^{-5}}$ $_{\pm 5.45 \cdot 10^{-6}}$ |
| MNIST (shifted) | N/A | $0.1938$ $_{\pm.0076}$ | $\mathbf{0.1552}$ $_{\pm.0280}$ |
| MNIST(rotated) | N/A | $0.2943$ $_{\pm.0109}$ | $\mathbf{0.0528}$ $_{\pm.0116}$ |
| Fashion-MNIST | N/A | $\mathbf{0.1636}$ $_{\pm.1665}$ | $0.2717$ $_{\pm.0864}$ |
| CIFAR-10 (test) | N/A | $\mathbf{0.0877}$ $_{\pm.1371}$ | $0.0983$ $_{\pm.0738}$ |
| CIFAR-10 (cyclic translation) | N/A | $\mathbf{0.1330}$ $_{\pm.0323}$ | $0.1646$ $_{\pm.0329}$ |
| CIFAR-10 (skewed) | N/A | $\mathbf{0.0877}$ $_{\pm.1534}$ | $0.0938$ $_{\pm.1371}$ |
| SVHN | N/A | $0.3916$ $_{\pm.1208}$ | $\mathbf{0.3611}$ $_{\pm.1191}$ |

Table 1: Average absolute difference of model uncertainty and std. of several runs ($N = 3$) between distillation models and ensemble of models. Knowledge distillation (Hinton et al., 2015) was not reported, as it does not offer uncertainty quantification. Lower is better.

et al., 2019). Model uncertainty, introduced by Depeweg et al. (2017); Malinin et al. (2019), is a measure of the spread or disagreement of an ensemble based on mutual information. A detailed description of MU and exemplary visualization can be found in Subsection 3.1. For classification we additionally measure classification accuracy and the Brier score. NLL has the disadvantage of over-emphasizing tail probabilities (Quinonero-Candela et al., 2005). In contrast, the Brier score, a proper scoring rule that takes into account both calibration and accuracy, is not as strongly skewed as NLL (Gneiting & Raftery, 2007). For a given input-output pair $(x^*, y^*)$ with $y^* \in [1, C]$ the Brier score is defined as $BS = \frac{1}{K} \sum_{c=1}^{C} \left( t^* - p(y = c|x^*) \right)^2$ where $t_c^* = 1$ if $c = y^*$, and 0 otherwise.

## 3.1 UNCERTAINTY QUANTIFICATION

We assess Hydra's ability to distill uncertainty metrics from an ensemble on classification tasks with MNIST and CIFAR-10. One way to quantify uncertainty is through model uncertainty (Depeweg et al., 2017; Malinin et al., 2019) which measures the spread or disagreement of an ensemble. Model uncertainty estimates the mutual information between the categorical label y and model parameters $\theta$. It can be expressed as the difference of the total uncertainty and the expected data uncertainty, where total uncertainty is the entropy of the expected predictive distribution and expected data uncertainty is the expected entropy of individual predictive distribution:

$$\underbrace{MU[y, \theta | x^*, \mathcal{D}]}_{\text{Model uncertainty}} = \underbrace{\mathcal{H}(\mathbb{E}_{p(\theta|D)}[p(y|x^*, \theta)])}_{\text{Total uncertainty}} - \underbrace{\mathbb{E}_{p(\theta|D)}[\mathcal{H}(p(y|x^*, \theta))]}_{\text{Expected data uncertainty}} \tag{6}$$

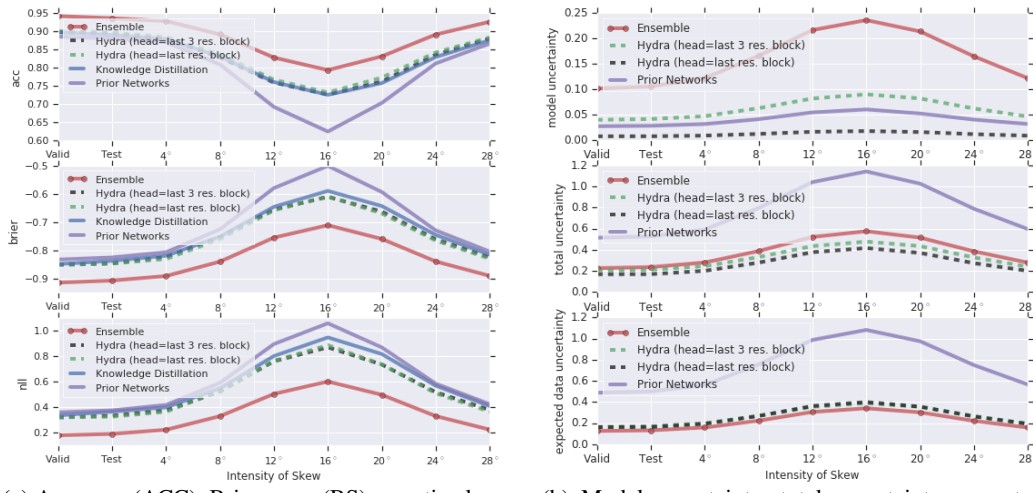

(a) Accuracy (ACC), Brier score (BS), negative log-likelihood (NLL)

(b) Model uncertainty, total uncertainty, expected data uncertainty

Figure 3: Model evaluation metrics plotted against intensity of distributional shift for CIFAR-10. The plots contain all distillation models as well as the original ensemble of models. We observe that Hydra consistently more accurately captures the uncertainty of the ensemble.

The total uncertainty will be high whenever the model is uncertain - both in regions of severe class overlap and out-of-domain. However, for out-of-distribution data the estimates of expected data uncertainty are poor, resulting in high model uncertainty. Figure 2 visualizes the model uncertainty and its decomposition for the spiral toy dataset, with Subfigures 2a and 2b showing results for the ensemble and Hydra respectively. The results show that Hydra successfully model uncertainty and its decomposition, though with a slight decrease in scale. We observe, as expected, a low model uncertainty where classes overlap due to both high total uncertainty and expected data uncertainty and high model uncertainty where at the border of in-domain and out-of-distribution data.

Further, we evaluate Hydra with respect to in-domain and out-of-distribution behavior for models trained on each MNIST and CIFAR10. We report the average absolute difference of model uncertainty as a measure how "close" the distillation method and ensemble are matched in Table 1. Hydra outperforms for MNIST test and shifted versions of MNIST, however, Hydra performs worse than Prior Network for several CIFAR10 datasets. Looking closer at the performance of one of the worse results, CIFAR10 (cyclic translation), we plotted all evaluation metrics against the intensity of skew. In order to rule out, that the worse performance is related to model capacity, we added the results from Hydra trained with a larger head of three residual blocks. As visualized in Figure 3, Hydra matches the behaviour of the ensemble the best in terms of accuracy, Brier score and NLL. With model uncertainty, Hydra with a larger head configuration seems to improve on overall performance compared to Prior Network. Inspecting the decomposition of model uncertainty, total uncertainty and expected data uncertainty, it seems like Hydra is more capable to "mimic" the behaviour the of ensemble. However, the uncertainty scales of Prior Networks is larger than the ones of both the ensemble and Hydra, leading to an overall better model uncertainty.

## 3.2 Test Performance on Common Classification and Regression Benchmarks

**Classification Performance on MNIST and CIFAR-10.** We investigate Hydra on two real image datasets, MNIST and CIFAR-10, using the setup described in Section 3. We report all metrics for MNIST in Table 2 and for CIFAR-10 in Table 3 as well as model capacity and efficiency in Table 5. For MNIST we can see that all distillation methods match the accuracy of the target ensemble, but Hydra outperforms both knowledge distillation and prior networks in terms of capturing the ensemble uncertainty, almost matching the ensemble predictive NLL (Hydra NLL 0.0465 matching ensemble NLL 0.0439) and Brier score (Hydra -0.9776 versus ensemble -0.9780). For the more challenging CIFAR-10 dataset all distillation methods approach but do not quite match the high accuracy of the target ensemble (0.9226 ensemble accuracy). Among distillation methods Hydra has the smallest gap in terms of accuracy. All distillation methods retain a gap in NLL performance compared to the ensemble, but Hydra again has a significantly smaller NLL (0.3179 NLL) compared to Prior networks (0.4392 NLL). In-distribution model uncertainty (MU) is comparable for both Prior

| Model | ACC $\uparrow$ | NLL $\downarrow$ | BS $\downarrow$ | MU |
|---|---|---|---|---|
| Ensemble ($M = 50$) | 0.9851 | 0.0439 | -0.9780 | $9.97 \cdot 10^{-6}$ |
| Prior Networks (Malinin et al. (2019)) | 0.9842 $_{\pm.0004}$ | 0.0521 $_{\pm.0007}$ | -0.9285 $_{\pm.0015}$ | 0.1158 $_{\pm.0013}$ |
| Knowledge distillation (Hinton et al. (2015)) | 0.9843 $_{\pm.0013}$ | 0.0497 $_{\pm.0050}$ | -0.9764 $_{\pm.0005}$ | N/A |
| Hydra (head $= [100, 100, 10]$) | **0.9857** $_{\pm.0004}$ | **0.0465** $_{\pm.0004}$ | **-0.9776** $_{\pm4.66 \cdot 10^{-6}}$ | $\mathbf{2.28 \cdot 10^{-5}}$ $_{\pm1.90 \cdot 10^{-6}}$ |

Table 2: Average test performance and std. over several runs ($n = 3$) for different baselines and Hydra for MNIST. For all models, classification accuracy (ACC), negative log-likelihood (NLL), Brier score (BS) and model uncertainty (MU) are reported. For #Params and FLOPs, we also report in brackets the percentage with respect to the ensemble. Bold numbers represent best performance with respect to the specific evaluation metric (columns) across distillation models (rows) except for MU, where we report the model uncertainty closest to the ensemble one.

| Model | ACC $\uparrow$ | NLL $\downarrow$ | BS $\downarrow$ | MU |
|---|---|---|---|---|
| Ensemble ($M = 50$) | 0.9226 | 0.2392 | -0.9033 | 0.1055 |
| Prior Networks (Malinin et al. (2019)) | 0.8731 $_{\pm.0035}$ | 0.4392 $_{\pm.0010}$ | -0.8231 $_{\pm.0015}$ | **0.0280** $_{\pm.0009}$ |
| Knowledge distillation (Hinton et al. (2015)) | 0.8933 $_{\pm.0020}$ | 0.3598 $_{\pm.0023}$ | -0.8373 $_{\pm.0004}$ | N/A |
| Hydra (head $=$ last res. block) | **0.8992** $_{\pm.0011}$ | **0.3179** $_{\pm.0080}$ | **-0.8468** $_{\pm.0030}$ | 0.0074 $_{\pm.0005}$ |

Table 3: Average test performance and std. over several runs ($n = 3$) for different baselines and Hydra for CIFAR-10. For all models, classification accuracy (ACC), negative log-likelihood (NLL), Brier score (BS) and Model Uuncertainty (MU) are reported. Bold numbers represent best performance with respect to the specific evaluation metric (columns) across distillation models (rows) except for MU, where we report the model uncertainty closest to the ensemble one.

Networks (0.0280) and Hydra (0.0074) but quite a bit smaller compared to target ensemble MU of 0.1055, meaning it is possible to improve uncertainty quantification in all distillation methods tested.

**Regression Performance on UCI Regression Datasets.** We trained both Knowledge Distillation (Hinton et al., 2015) and Hydra on standard regression UCI datasets shown in Table 4. Here Prior Networks are not applicable because for the case of probabilistic regression we cannot take averages of distributions.[3] For regression Hydra outperforms knowledge distillation in terms of predictive performance (NLL) because Hydra produces a more flexible output in the form of a Gaussian mixture model with one Gaussian component per head, whereas Knowledge Distillation can produce only a single Gaussian component.

## 4  RELATED WORK

We now review related work in both distillation and multi-headed neural architectures.

**Distillation.** In (Hinton et al., 2015), a "student" network—i.e. the outcome of the distillation process—is trained to match the *average* predictions of the "teacher" network(s). This methodology has been later successfully applied to the distillation of Bayesian ensembles (Balan et al., 2015) where ensemble members correspond to a Monte Carlo approximation to the posterior distribution.[4] A parallel line of research has focused on *co-distillation*, also sometimes referred to as *online distillation* to further reduce the overall training cost (Zhang et al., 2018; Anil et al., 2018; Lan et al., 2018). In this setting, both the student and teacher networks are learned simultaneously and this form of mutual training acts as a regularization mechanism. Also, distillation has been recently the topic of theoretical analysis to better explain its empirical success (Lopez-Paz et al., 2015; Phuong & Lampert, 2019).

---

[3]Formally, the average of two Gaussian densities is no longer Gaussian in general.

[4]Note that, in this paper, we focus primarily on instances of distillation where the supervision occurs at the level of the predictive probabilities, although other strategies, e.g., based on the feature representations in the networks (Ba & Caruana, 2014; Romero et al., 2014), have been proposed.

| Dataset | Ensemble | Prior Network (Malinin et al., 2019) | Knowledge distillation (Hinton et al. (2015)) | Hydra (head = [10, 1]) |
|---|---|---|---|---|
| bost | 2.3780 | N/A | $2.3893_{\pm.0089}$ | $\mathbf{2.3805}_{\pm.0084}$ |
| concr | 3.0585 | N/A | $3.1231_{\pm.0054}$ | $\mathbf{3.0982}_{\pm.0065}$ |
| ener | 1.2756 | N/A | $1.5236_{\pm.0277}$ | $\mathbf{1.4402}_{\pm.0055}$ |
| kin8 | -1.2977 | N/A | $-1.2343_{\pm.0014}$ | $\mathbf{-1.2555}_{\pm.0034}$ |
| nava | -7.5983 | N/A | $-6.4340_{\pm.2710}$ | $\mathbf{-7.1987}_{\pm.1193}$ |
| powe | 2.8861 | N/A | $2.8940_{\pm.0007}$ | $\mathbf{2.8921}_{\pm.0007}$ |
| prot | 2.8272 | N/A | $2.8970_{\pm.0007}$ | $\mathbf{2.8829}_{\pm.0064}$ |
| wine | 0.9111 | N/A | $\mathbf{0.9112}_{\pm.0012}$ | $0.9113_{\pm.0010}$ |
| yach | -0.1640 | N/A | $0.3837_{\pm.0174}$ | $\mathbf{0.3489}_{\pm0.0136}$ |

Table 4: UCI regression benchmark (Dua & Taniskidou, 2017). Average test negative log-likelihood (NLL) and std. over several runs ($n = 3$) of the 9 different datasets considered. As Prior Networks (Malinin et al., 2019) cannot be applied to regression tasks, we denote this with "N/A".

| Model | MNIST | | CIFAR-10 | |
|---|---|---|---|---|
| | #Params $\downarrow$ | FLOPs $\downarrow$ | #Params $\downarrow$ | FLOPs $\downarrow$ |
| Ensemble ($M = 50$) | 9,960,500 | $9.0 \cdot 10^7$ | 13,722,100 | $4.09 \cdot 10^{10}$ |
| Prior Networks (Malinin et al. (2019)) | **199,210 [2%]** | $\mathbf{1.8 \cdot 10^6}$ **[2%]** | **274,442 [2%]** | $\mathbf{8.18 \cdot 10^8}$ **[2%]** |
| Knowledge distillation (Hinton et al. (2015)) | **199,210 [2%]** | $\mathbf{1.8 \cdot 10^6}$ **[2%]** | **274,442 [2%]** | $\mathbf{8.18 \cdot 10^8}$ **[2%]** |
| Hydra (head = [100, 100, 10]) | 1,757,700 [17.6%] | $2.5 \cdot 10^7$ [27.7%] | - | - |
| Hydra (head=last res.block) | - | - | 3,950,324 [28.7%] | $5.45 \cdot 10^9$ [13.3%] |

Table 5: Model capacity and efficiency: We report number of model parameters (#Params) and number of floating point operations (FLOPs) for the ensemble and all distillation models. We also report in brackets the percentage with respect to the ensemble.

Closest to our approach is the work of Lan et al. (2018). Their method consists in training multiple student models whose combined predictions induce an ensemble teacher model. While we share conceptual similarities with their work, we depart from their formulations in several ways. First, we focus on the offline ensemble setting (Hinton et al., 2015) where we start from a pre-defined ensemble whose training may be difficult to replicate inside a co-distillation process. Second, our approach follows a different goal: we consider multiple branches, or *heads* in our terminology, to *individually match the behavior of each teacher model*. We do so to preserve the diversity of the teacher ensemble which is, for instance, essential in out-of-distribution tasks (Lakshminarayanan et al., 2017; Ovadia et al., 2019). Third, our methodology has a conceivably simpler design, as reflected by our single-component objective function—the average KL divergence between each head and corresponding teacher model—and the absence of a learned gating mechanism to linearly combine the logits of the student models (Lan et al., 2018).

**Multi-headed architectures.** This type of architecture can be motivated by several benefits, e.g., a reduction in memory footprint due to the sharing of parameters, a speed-up of the training process, a regularization effect due to the introduction of auxiliary training objectives, or the transfer of information across different tasks. As discussed in detail by Song & Chai (2018), there exist multiple strategies to define a *body* of shared parameters together with *heads*, e.g. hierarchical sharing pattern. In this work, we concentrate on simple multi-headed architectures, where the heads will correspond to either the last-layer of the original network or small extensions thereof. While multi-headed architectures were used for online distillation in (Lan et al., 2018), we reiterate that our goal is different from this previous work in that we exploit the multiple heads to match and mirror the individual members of a pre-defined teacher ensemble.

## 5 CONCLUSION

We presented *Hydra*, a simple and effective approach to distillation for ensemble models. Hydra preserves diversity in ensemble member predictions and we have demonstrated on standard models that capturing this information translates into improved performance and better uncertainty quantification. While Hydra improves on previous approaches we believe that we can further improve

distillation performance by leveraging techniques from fields studying sets of related learning such as meta-learning and domain adaptation.

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
