# OpenReview forum: "Hydra: Preserving Ensemble Diversity for Model Distillation"
_ICLR.cc/2020/Conference — Reject_

### Official Review · AnonReviewer3 · 2019-10-16
**Official Blind Review #3**

**Rating:** 3

**Review:**

Summary & Pros
- This paper proposes a simple yet effective distillation scheme from an ensemble of independent models to a multi-head architecture for preserving the diversity of the ensemble.
- The proposed scheme provides the same advantages of the ensemble in terms of uncertainty estimation and predictive performance, but it is computationally efficient compared to the ensemble.
- This paper experimentally shows that multi-head architecture performs well on MNIST and CIFAR-10 in terms of accuracy and uncertainty.

Concerns #1: Novelty of the proposed method
- The multi-head architectures have been widely used in various settings, especially multi-task learning. As the authors mentioned, it also used for online distillation [1]. Although its goal is different from this paper, just applying such multi-head architectures seems to be incremental.

Concerns #2: Insufficient experiments
- To evaluate OOD detection quality, ID/OOD datasets should be stated and various metrics (e.g., AUROC) should be measured like other literature, e.g., Table 2 in [2]. Such OOD detection quality is important to evaluate the quality of uncertainty estimation.
- This paper provides experiments on only small-sized 10-class datasets, MNIST and CIFAR10. To verify the effectiveness of the proposed distillation method, other large-sized datasets should be tested, e.g., CIFAR-100, ImageNet.
- There is no ablation study on the effect of the number of size of heads in Hydra. To achieve similar performance to the ensemble, how many heads are required?

Concerns #3: Week efficiency
- As reported in Table 5, in the case of CIFAR10, Hydra has 14x more parameters and 6x more FLOPs. Despite such a large number of parameters, the performance gain seems to be incremental.
- A comparison with an ensemble with M=14 models should be tested because this ensemble has the same number of parameters compared to Hydra with M=50 heads. I think it might achieve good performance on the evaluation metrics.

Concerns #4: Incremental improvements
- Accuracy gain is too marginal even Hydra uses 14x more parameters.
- ONE [1] might be a stronger baseline because ONE achieves 94% accuracy on CIFAR-10 using ResNet32 with only 2~3 heads while Hydra achieves only 90% even it uses 50 heads. Moreover, since ONE has multiple heads, uncertainty estimation is also available. So it should be compared with the proposed method.
- In OOD detection tasks, Hydra underperforms Prior Networks on 5 of 8 datasets (note that PN (2.60) is better than Hydra (3.11) in the case of MNIST (test)). To overcome this gap, the proposed method requires more parameters.

[1] Zhu, Xiatian, and Shaogang Gong. "Knowledge Distillation by On-the-Fly Native Ensemble." Advances in Neural Information Processing Systems. 2018.
[2] Andrey Malinin and Mark Gales. "Predictive Uncertainty Estimation via Prior Networks." Advances in Neural Information Processing Systems. 2018.


**Experience Assessment:**

I have read many papers in this area.

**Review Assessment: Checking Correctness Of Derivations And Theory:**

I assessed the sensibility of the derivations and theory.

**Review Assessment: Checking Correctness Of Experiments:**

I assessed the sensibility of the experiments.

**Review Assessment: Thoroughness In Paper Reading:**

I read the paper at least twice and used my best judgement in assessing the paper.

---

> ### Author Response · Authors · 2019-11-15
> **Response to AnonReviewer3 (part I)**
>
> [R3.1] This paper proposes a simple yet effective distillation scheme from an ensemble of independent models to a multi-head architecture for preserving the diversity of the ensemble. [...]
> [Response] We would like to thank the reviewers for the detailed feedback and hope that the responses helps clarify major concerns.
>
> [R3.2] The multi-head architectures have been widely used in various settings, especially multi-task learning. As the authors mentioned, it also used for online distillation [1]. Although its goal is different from this paper, just applying such multi-head architectures seems to be incremental.
> [Response] We argue that although multi-headed architectures have been explored, for the case of offline distillation they have not been. Especially in our work, we focus on multi-headed architecture for uncertainty estimation. This is a direction which has not been proposed to the best of our knowledge. Further, our proposed objective is simple, the average KL divergence between each head and corresponding teacher model, and versatile as it can be applied to both classification and regression.
>
> [R3.3] To evaluate OOD detection quality, ID/OOD datasets should be stated and various metrics (e.g., AUROC) [...]
> In our work, we focus on the quality of uncertainty preservation and thus report strictly proper scoring rules (Gneiting & Raftery, 2007) following metrics proposed by Ovadia et al., 2019 (Brier score) and the follow-up work of Prior Networks of Malinin et al., 2019 (model uncertainty). We believe that both metrics are appropriate ID/OOD evaluation as this has been used in the works aforementioned. However, we appreciate this suggestion and will add this to the paper for the camera-ready version.
>
> [R3.4] ]This paper provides experiments on only small-sized 10-class datasets, MNIST and CIFAR10. To verify the effectiveness of the proposed distillation method, other large-sized datasets should be tested, e.g., CIFAR-100, ImageNet.
> [Response] We have run extended experiments on CIFAR-100. We used an ensemble of 10 Resnet20 members which were trained separately for distillation. For both Prior Networks and Knowledge distillation, we used Resnet20 with varying number of linear layers ([100, 100, 100], [300, 300, 100], [800, 800, 100]) after the residual blocks of Resnet20. For Hydra we used 10 heads to distill 10 ensemble members, each head had varying number of linear layers ([100, 100, 100], [300, 300, 100], [800, 800, 100]) and shared a Resnet20 as body. We optimized over a wide range of hyperparameters for both Hydra and baseline models (knowledge distillation, Prior Networks) and can show improvement for both accuracy and NLL (see table below). We also show that the test model uncertainty is closer to the ones of the ensemble.
>
> +-------------------------------------------------------------+----------+----------+-----------+
> |                    Methods                                            |  NLL    |  ACC    |   MU     |
> +-------------------------------------------------------------+----------+----------+-----------+
> | Individual NN                                                      | 1.4282 | 0.6713 | -           |
> | Ensemble (N=10)                                                | 0.8764 | 0.7585 | 0.3709 |
> | Knowledge distillation (Hinton et al. (2015)) | 0.9637 | 0.7205 | N/A      |
> | Prior Networks (Malinin et al. (2019))             | 3.0705 | 0.6638 | 0.0088 |
> | Hydra                                                                    | 0.9421 | 0.7314 | 0.1259 |
> +-------------------------------------------------------------+-----------+----------+----------+
>
> [R3.5] There is no ablation study on the effect of the number of size of heads in Hydra. To achieve similar performance to the ensemble, how many heads are required?
> We assume as many heads as we have ensemble members. Since we have for both MNIST and CIFAR10 ensembles of 50 members, the number of heads for Hydra is by default set to 50. Therefore, an ablation study would not give much insight in this case.
>
> [R3.6] As reported in Table 5, in the case of CIFAR10, Hydra has 14x more parameters and 6x more FLOPs. [...] A comparison with an ensemble with M=14 models should be tested because this ensemble has the same number of parameters compared to Hydra with M=50 heads. [...]
> [Response] We ran out of time during the rebuttal phase to conduct this additional study for CIFAR-10. However, as part of the new experiments we conducted for CIFAR-100, we obtained good performance while using just dense layers for the heads. Comparatively, an ensemble with M=10 obtains the following performance of 73.14 accuracy and a negative log-likelihood of 0.9421.

---

> > ### Author Response · Authors · 2019-11-15
> > **Response to AnonReviewer3 (part II)**
> >
> > [R3.7] ONE [1] might be a stronger baseline [..]. Moreover, since ONE has multiple heads, uncertainty estimation is also available. [...]
> > [Response] [1] only uses 3 heads, i.e. they distill an ensemble of three members with the proposed ONE. We used an ensemble of 50 members, therefore we cannot compare directly to the results reported in [1]. Due to time constraints of the rebuttal, we will add the comparison for the camera-ready version.
> >
> > [R3.8] In OOD detection tasks, Hydra underperforms Prior Networks on 5 of 8 datasets (note that PN (2.60) is better than Hydra (3.11) in the case of MNIST (test)). To overcome this gap, the proposed method requires more parameters.
> > [Response] Thank you for this suggestion. We will explore different kinds of architecture and ways of weight sharing in future work.

---

### Official Review · AnonReviewer2 · 2019-10-22
**Official Blind Review #2**

**Rating:** 6

**Review:**

Overview:
This work introduces a new method for ensemble distillation. The problem of making better ensemble distillation methods seems relevant as ensembles are still one of the best ways to estimate uncertainty in practice (although see concerns below). The method itself is a simple extension of earlier “prior networks”: the original method suggested to fit a single network to mimick a distribution produce by given ensemble, and here authors suggest to use multi-head (one head per individual ensemble member) in order to better capture the ensemble diversity.
Authors report results on multiple relatively standard benchmarks (MNIST, CIFAR, etc), and seem to outperform the baseline by a small margin. The choice of baselines is reasonable.

Writing:
The paper is well-written, illustrations are good.

Decision:
The method itself is very easy to implement, and does seem to outperform the baseline (prior networks). However, I am a bit concerned that the method itself seems like a trivial extension of the prior work, and does not really provide much addition insight. In addition, the results are reported on a set of small-scale benchmarks and seem incremental: it can be OK, but it would be really great to see a somewhat more realistic application.
Thus, I am on the fence with this one, but generally positive about this work, thus “weak accept” rating.

Questions / concerns:
* I honestly do not see the point on having an additional column in tables if all the values are N/A.
* The names in Table 4 are mixed up.
* Arguably, a lot of applications that would actually rely on uncertainty estimation might require online training of some sort. This means that in those scenarios one does not actually have access to a pre-trained ensemble. I understand that this might not be the main focus of this work, but it seems like a major limitation of “distillation” approaches in general, which should / could be addressed in some way?

<update>
Thanks for a detailed answer. I am not very convinced by the argument about online-vs-offline training, but I do not see a reason to decrease my rating. I can see this work useful in practice.
</update>


**Experience Assessment:**

I have read many papers in this area.

**Review Assessment: Checking Correctness Of Derivations And Theory:**

I assessed the sensibility of the derivations and theory.

**Review Assessment: Checking Correctness Of Experiments:**

I carefully checked the experiments.

**Review Assessment: Thoroughness In Paper Reading:**

I read the paper thoroughly.

---

> ### Author Response · Authors · 2019-11-15
> **Response to AnonReviewer2**
>
> [R2.1] This work introduces a new method for ensemble distillation. The problem of making better ensemble distillation methods seems relevant as ensembles are still one of the best ways to estimate uncertainty in practice (although see concerns below). [...] The paper is well-written, illustrations are good.
> [Response] We would like to thank the reviewer for the valuable feedback. We hope that the additional experiments and the responses helps clarifying concerns.
>
> [R2.2] The method itself is very easy to implement, and does seem to outperform the baseline (prior networks). However, I am a bit concerned that the method itself seems like a trivial extension of the prior work, and does not really provide much addition insight.
> [Response] The focus of our proposed work is knowledge distillation which preserves model uncertainty. We do this by using a multi-headed architecture, an objective which has not been suggested for model uncertainty preservation before. We do believe that this simple objective is versatile while able to preserve uncertainty.
>
> [R2.3] In addition, the results are reported on a set of small-scale benchmarks and seem incremental: it can be OK, but it would be really great to see a somewhat more realistic application.
> [Response] We followed your advice and performed evaluation on CIFAR100.  We used an ensemble of 10 Resnet20 members which were trained separately for distillation. For both Prior Networks and Knowledge distillation, we used Resnet20 with varying number of linear layers after the residual blocks of Resnet20. For Hydra we used 10 heads to distill 10 ensemble members, each head had varying number of linear layers and shared a Resnet20 as body. We optimized over a wide range of hyperparameters for both Hydra and baseline models (knowledge distillation, Prior Networks). All details can be found in the general comment above. In the table below, we report results with Hydra and all baseline/state-of-the-art models. We show improvement for both accuracy and NLL. We also show that the test model uncertainty is closer to the ones of the ensemble.
>
> +-------------------------------------------------------------+----------+----------+-----------+
> |                    Methods                                            |  NLL    |  ACC    |   MU     |
> +-------------------------------------------------------------+----------+----------+-----------+
> | Individual NN                                                      | 1.4282 | 0.6713 | -           |
> | Ensemble (N=10)                                                | 0.8764 | 0.7585 | 0.3709 |
> | Knowledge distillation (Hinton et al. (2015)) | 0.9637 | 0.7205 | N/A      |
> | Prior Networks (Malinin et al. (2019))             | 3.0705 | 0.6638 | 0.0088 |
> | Hydra                                                                    | 0.9421 | 0.7314 | 0.1259 |
> +-------------------------------------------------------------+-----------+----------+----------+
>
> [R2.4] I honestly do not see the point on having an additional column in tables if all the values are N/A.
> [Response] This is there to emphasize the limitations of Knowledge distillation and Prior networks. Knowledge distillation is not able to output any uncertainty estimates, and Prior Networks are usually inferior to Knowledge distillation and can only be applied to classification tasks. Our method improves on both classification and regression tasks and offers uncertainty estimation. We have added a brief explanation in the table caption.
>
> [R2.5] The names in Table 4 are mixed up.
> [Response] Thank you for spotting this. We have fixed this in the current revision.
>
> [R2.6] Arguably, a lot of applications that would actually rely on uncertainty estimation might require online training of some sort. [...] I understand that this might not be the main focus of this work, but it seems like a major limitation of “distillation” approaches in general, which should / could be addressed in some way?
> [Response] We would argue the other way around. It is actually more difficult to adapt different training regimes to support co-training of teacher and student models. For example, Lan et al. propose a multi-branch online distillation. For their model, all teacher models need to be trained concurrently to the student model. This requires a high number of GPUs for all teacher models and student model to be trained efficiently. We are not confident that this can scale to an ensemble of 50 members - the number of ensemble members we used. Lan et al. only show their model on an ensemble of three members. Further, in cases where training is not straightforward, e.g. Bayesian Neural Networks, recurrent models, Progressive Growing GAN, designing a general purpose online training might be difficult. Nonetheless, we believe for the models proposed an iterative manner of training both teacher and student may be possible. However, we leave exploring this direction to future work.

---

### Official Review · AnonReviewer1 · 2019-10-23
**Official Blind Review #1**

**Rating:** 1

**Review:**

The paper proposes to distill the predictions of an ensemble with a multi-headed network, with as many heads as members in the original ensemble. Distillation proceeds by minimizing the KL divergence between the predictions of each ensemble member with the corresponding head in the student network. Experiments illustrate that the multi-headed architecture approximates the ensemble marginally better than approaches that use a network with a single head.

The paper presents a straightforward idea and fairly unsurprising results —  a multi-headed architecture with each head matching an ensemble member more faithfully represents the original ensemble. This improved fidelity, however, comes at the cost of increased computation and storage requirements (which scale linearly with the size of the ensemble). It is unclear that the marginal improvements demonstrated justify the increased cost and how this approach would scale to larger ensembles. The paper would have been more interesting if the authors had managed to demonstrated significant improvements over competitors on not toy (MNIST / CIFAR) problems. Unfortunately, this is not the case and the fact that similar ideas (Lan et al.) have been proposed in the past (which the authors, to their credit, cite) leads me to recommend a rejection.

**Experience Assessment:**

I have published one or two papers in this area.

**Review Assessment: Checking Correctness Of Derivations And Theory:**

N/A

**Review Assessment: Checking Correctness Of Experiments:**

I assessed the sensibility of the experiments.

**Review Assessment: Thoroughness In Paper Reading:**

I read the paper thoroughly.

---

> ### Author Response · Authors · 2019-11-15
> **Response to AnonReviewer1**
>
> [R1.1] The paper proposes to distill the predictions of an ensemble with a multi-headed network, [...] The paper presents a straightforward idea and fairly unsurprising results [...]
> [Response] We would like to thank the reviewer for the feedback.
> [R1.2] It is unclear that the marginal improvements demonstrated justify the increased cost and how this approach would scale to larger ensembles. The paper would have been more interesting if the authors had managed to demonstrate significant improvements over competitors on not toy (MNIST / CIFAR) problems.
> [Response] In order to demonstrate improvements on “not toy” problems, we applied Hydra and baseline models to CIFAR-100. Although the complexity of the model remains the same (Resnet-20), the complexity of the task increases. In particular, with the higher number of classes, the improvement of the ensemble compared with a single model is more pronounced (around xx % CIFAR-10 and around 10% improvement for CIFAR-100), which makes the task of properly distilling the ensemble even more relevant and critical. For CIFAR-100, we also use additional linear layers as Hydra heads. We optimized over a wide range of hyperparameters for both Hydra and baseline models (knowledge distillation, Prior Networks) and can show improvement for both accuracy and NLL (see table below). We also show that the test model uncertainty is closer to the ones of the ensemble, Prior Network’s model uncertainty is much lower than the one of the original ensemble and knowledge distillation cannot be used for model uncertainty estimation.
>
> +-------------------------------------------------------------+----------+----------+-----------+
> |                    Methods                                            |  NLL    |  ACC    |   MU     |
> +-------------------------------------------------------------+----------+----------+-----------+
> | Individual NN                                                      | 1.4282 | 0.6713 | -           |
> | Ensemble (N=10)                                                | 0.8764 | 0.7585 | 0.3709 |
> | Knowledge distillation (Hinton et al. (2015)) | 0.9637 | 0.7205 | N/A      |
> | Prior Networks (Malinin et al. (2019))             | 3.0705 | 0.6638 | 0.0088 |
> | Hydra                                                                    | 0.9421 | 0.7314 | 0.1259 |
> +-------------------------------------------------------------+-----------+----------+----------+
>
> [R1.3] Unfortunately, this is not the case and the fact that similar ideas (Lan et al.) have been proposed in the past (which the authors, to their credit, cite) leads me to recommend a rejection.
> [Response] We would like to refer to our related work for a detailed comparison to the work of Lan et al. Although we share conceptual similarities with their work, our work differs from theirs in several ways. We focus on offline distillation which can be used for any kind of ensemble and any number of ensemble members, even ones whose training may be difficult to replicate. For instance, Bayesian ensembles might be difficult to be trained within the co-distillation process of Lan et al. A further benefit of Hydra is its simple design which is reflected in our single-component objective function. Hydra does not need to learn an additional gating mechanism to linearly combine the logits of the student models. Lan et al. only show their proposed model for a three-branch model and it is unclear how effective and efficient the co-distillation is with 50 branches (which is the setting that we used).

---

### Author Response · Authors · 2019-11-15
**General comments to the AC and the reviewers**

First of all, we would like to thank the reviewers for their feedback. In the following, we summarize and address major concerns raised by all reviewers:
1) Lack of novelty: All reviewers criticize the lack of novelty as multi-headed architectures have been explored in various areas of machine learning. Further, reviewer 1 and 3 note that a related work (Lan et al, 2018) has already proposed a multi-branch architecture for distillation of ensembles.
[Response] We would like to emphasize that our focus was to introduce a general and simple method for distillation which preserves model uncertainty. Using a multi-headed architecture has not been proposed so far. Our method can be applied to both classification and regression tasks (which baselines like Prior Networks cannot) while being able to preserve model uncertainty (which popular method like knowledge distillation cannot). In comparison to (Lan et al., 2018), our method is thought to be used for offline distillation whereas Lan et al. proposed a method for online distillation, also known as co-distillation. We argue that co-distillation training is difficult to adapt to different kinds of model (Bayesian neural networks, progressive growing models, recurrent models) and difficult to scale as all ensemble models are trained simultaneously with the student model during distillation. In fact, Lan et al. only show a distillation of an ensemble of three members. Our method is able to scale, and we have shown this with ensemble of 50 members. Further, our methods can be easily applied to any kind of model as we do not rely on training the teacher models at the same time as the student model.
2) Lack of large-scale experiments: All reviewers were concerned that current evaluation does not include large-scale datasets and architecture, thus, are concerned how scalable the method is and whether improvements are also made in such large-scale settings.
[Response] As advised by all reviewers, we conducted further experiments with CIFAR-100. Although the model complexity stays the same (we used a Resnet20), the task complexity increases. Due to the time constraint, we only experiment with MLP heads in order to be able to do extensive hyperparameter for fair comparisons between Hydra and models used for comparison. As shown in the table below, Hydra improves Knowledge distillation and Prior Networks for both negative log-likelihood and accuracy. Further, we are able to have higher model uncertainty estimates than Prior Networks, and also are closer to what the ensemble outputs. We have not yet included the results to the paper, as we would like to run more experiments to include larger ensembles and the last residual blocks as Hydra heads. These results will then be added to the final camera-ready version.

+-------------------------------------------------------------+----------+----------+-----------+
|                    Methods                                            |  NLL    |  ACC    |   MU     |
+-------------------------------------------------------------+----------+----------+-----------+
| Individual NN                                                      | 1.4282 | 0.6713 | -           |
| Ensemble (N=10)                                                | 0.8764 | 0.7585 | 0.3709 |
| Knowledge distillation (Hinton et al. (2015)) | 0.9637 | 0.7205 | N/A      |
| Prior Networks (Malinin et al. (2019))             | 3.0705 | 0.6638 | 0.0088 |
| Hydra                                                                    | 0.9421 | 0.7314 | 0.1259 |
+-------------------------------------------------------------+-----------+----------+----------+

The settings for the experiments are given below:

Number of ensemble: 10
Number of layers for heads: [300, 300, 100], [800, 800, 100], [100, 100, 100] (also used for Knowledge distillation and Prior Networks as last layers)
Weight decay: [1e-2, 1e-3, 1e-4, 1e-5, 1e-6, 1e-7, 1e-8]
temperature: [1., 2.5, 5., 7.5, 10.]
Dropout rate: [0.0, 0.1, 0.25, 0.5, 0.75, 0.9]

Detailed comments to all reviews are below as replies to each review.

---

### Decision · Program_Chairs · 2019-12-19

**Decision:**

Reject

**Comment:**

This work introduces a simple and effective method for ensemble distillation. The method is a simple extension of earlier “prior networks”: it differs in which, instead of fitting a single network to mimic a distribution produced by the ensemble, this work suggests to use multi-head (one head per individual ensemble member) in order to better capture the ensemble diversity. This paper experimentally shows that multi-head architecture performs well on MNIST and CIFAR-10 (they added CIFAR-100 in the revised version) in terms of accuracy and uncertainty.

While the method is effective and the experiments on CIFAR-100 (a harder task) improved the paper, the reviewers (myself included) pointed out in the discussion phase that the limited novelty remains a major weakness. The proposed method seems like a trivial extension of the prior work, and does not provide much additional insight. To remedy this shortcoming, I suggest the authors provide extensive experimental supports including various datasets and ablation studies.

Another concern mentioned in the discussion is the fact that these small improvements are in spite of the fact that the proposed method ends up using many more parameters than the baselines. Including and comparing different model sizes in a full fledged experimental evaluation would better convey the trade-offs of the proposed approach.